# The Hospital Patient Safety Culture Survey: Reform of Analysis and Visualization Methods

**DOI:** 10.3390/ijerph16193624

**Published:** 2019-09-27

**Authors:** Heon-Jae Jeong, Wui-Chiang Lee, Hsun-Hsiang Liao, Feng-Yuan Chu, Tzeng-Ji Chen, Pa-Chun Wang

**Affiliations:** 1Joint Commission of Taiwan, No. 31, Sec. 2, Sanmin Rd., Banqiao Dist., New Taipei City 220, Taiwan; hj9571@gmail.com (H.-J.J.); shing@jct.org.tw (H.-H.L.); pachun.wang@jct.org.tw (P.-C.W.); 2The Care Quality Research Group, 174 Toegye, Chuncheon, Gangwon 24450, Korea; 3Department of Medical Affairs and Planning, Taipei Veterans General Hospital, No. 201, Sec. 2, Shipai Rd., Beitou Dist., Taipei City 112, Taiwan; 4Institute of Hospital and Healthcare Administration, National Yang-Ming University School of Medicine, No. 155, Sec. 2, Linong St., Beitou Dist., Taipei City 112, Taiwan; tjchen@vghtpe.gov.tw; 5Department of Family Medicine, Taipei Veterans General Hospital, Yuanshan & Su-Ao Branch, No. 386, Rongguang Rd., Yuanshan Township, Yilan County 264, Taiwan; steven2259898@gmail.com; 6Department of Family Medicine, Taipei Veterans General Hospital, No. 201, Sec. 2, Shi-Pai Rd., Beitou Dist., Taipei 112, Taiwan

**Keywords:** percent agreement, safety culture, safety attitudes, survey, Taiwan

## Abstract

Understanding the topography of hospital safety culture is vital for developing, implementing, and monitoring the effectiveness of tailored safety programs. Since 2009, the Chinese version of the Safety Attitudes Questionnaire (SAQ-C) has been introduced and administered to providers in many Taiwanese hospitals. The percentage of SAQ respondents who demonstrate attitudinal agreement within each of the SAQ domains, the percent agreement (PA) score, is used worldwide as the main parameter of safety culture surveys. However, several limitations within PA scoring have been identified. Our study sought to improve scoring methodology and develop a new graph layout for cultural topography presentation. A total of 37,163 responses to a national SAQ-C administration involving 200 Taiwan hospitals were retrospectively analyzed. To understand the central tendency and spread of safety culture scores across all participating hospitals, the median and interquartile range (IQR) of individual respondent’s SAQ-C scores by its domain were calculated, plotted, and named “Jeong & Lee plot”. Because of space limitation, we showed teamwork domain only. Study results denote limitations in the current PA scoring scheme, suggest SAQ analysis modification, and introduce a visualization graph layout that can provide richer information about safety culture dissemination than that available from currently utilized tools.

## 1. Introduction

Since the groundbreaking 1999 Institute of Medicine report, “To Err is Human: Building a Safer Health System” was published, patient safety has taken central stage in healthcare, leading to global development of safety improvement programs [1,2,3,4]. These programs target specific safety issues (e.g., wrong site surgeries, patient falls) and are connected to the development of organizational safety culture [4,5,6,7,8]. Understanding the topography of safety culture is vital for developing and implementing tailored safety programs and monitoring their effectiveness. 

For a decade, most Taiwanese hospitals have implemented The Safety Attitudes Questionnaire (SAQ). Dr. Bryan Sexton developed the original version of SAQ [9]. In 2007 it was introduced to Taiwan and translated into the Chinese language version (SAQ-C) by Lee et al. under the developer’s authorization [10]. After validation by large-scale studies [10,11], the Joint Commission of Taiwan (JCT) in 2009 promoted the survey to hospitals across the country as the Taiwan Patient Safety Culture (TPSC) survey [11]. Participating hospitals received feedback on their performance and that of their peers. As a result, the annual safety culture survey has emerged as an essential tool for continuous hospital safety improvement. The SAQ-C instrument has also been adopted in China, Hong Kong, and Macau [12,13].

The SAQ-C consists of 32 items measuring provider attitudes in five domains: teamwork climate (six items), safety climate (seven items), job satisfaction (five items), perception of management (ten items), and working conditions (four items). A 5-point Likert scale (1 = disagree strongly, 2 = disagree slightly, 3 = neutral, 4 = agree slightly, 5 = agree strongly) measures the level of agreement with each item. After survey administration, raw scores are converted into a 0–100 scale with 25 as the gap between the nearest scores. For each respondent, the average of the five scores is then calculated (0–100). The next step is to dichotomize mean scores above or equal to 75, or below 75. The former indicates the percentage of positive attitudinal responses in each domain and the latter delineates the rate of negative attitudinal reactions. Dividing the number of agreeing respondents by the number of total respondents yields the percentage agreement (PA). The result is then multiplied by 100 to discern the percentage scale [9,11].

Since its creation, PA and its visual presentation, usually in the form of a densely-populated horizontal table, has been utilized to show cluster-level scores [9]. PA calculation is simple as it does not entail complex statistical techniques, and the presentation graph is easy to understand. Nevertheless, there are concerns about this popular scoring scheme. First, the dichotomization method has an impact. For example, if a survey respondent assigned “agree slightly” to all six items within the teamwork climate (TC) domain, her average score would be 75 and thus categorized as positive agreement. However, if she assigned any item as “neutral” in the next survey, her mean score would be 70, indicating negative agreement. Second, a change in a person’s agreement status can be double counted. Following the example above, when the respondent changed “agreement” to “disagreement” within the domain, there was a simultaneous increase in disagreement and agreement. The influence on PA was inflated because of the unusual ratio statistics. Third, the distribution pattern was not taken into account. The current PA scoring scheme addresses the group mean but ignores the respondents scoring below 75 and their changed safety-related attitudes. In one example, the PA of a hospital would not change even if the positive survey responses of its workforce indicated an improved score (changing from 25 to 70) in safety-related attitudes. As a result, the performance of safety initiatives cannot be precisely assessed, a flaw which can mislead the management in decision-making. 

Given the limitations of the current PA scoring approach, this study sought to develop a new SAQ-C analysis methodology that can provide richer information about the development of safety culture and related changes in the five domains.

## 2. Materials and Methods 

### 2.1. Data Sources

Survey data collected from 200 Taiwan hospitals from 31 May 2008, to 30 June 2008, was used in this study. We calibrated the first-year data because of the possibility of item parameter drift (IPD) [14], where multiple survey takers have different item parameters (i.e., different reaction patterns). Because of the anonymous nature of a survey like SAQ-C, in which a respondent cannot be tracked down, the best way to eliminate IPD risk was to calibrate the uncontaminated first-year dataset. Study results could then be compared to the previous study using the same dataset [11]. Given space limitations, this study examined the TC survey data as one example among the five SAQ-C domains.

### 2.2. Understanding PA’s Limitations

This study calculated the mean PA score for each hospital’s TC domain. A traditional plot for the PA was also generated, showing the difference in the hospital’s PA distribution. We selected hospitals with the same mean PA score in the TC domain for further comparison. A kernel density estimate (KDE) plot for each hospital compared the differences between the distribution and shape. 

### 2.3. Choosing and Calculating Cluster-Level Statistics

To address the central tendency and spread of safety culture scores across of each participating hospital, the median and interquartile range (IQR) of each individual respondent’s scores to the TC domain was calculated, plotted, and named “Jeong & Lee plot”. The mean and standard deviation were not used because they did not function well when describing highly abnormal distributed data or data with outliers. The new safety grid plot could concurrently depict both central tendencies and the spread of safety culture scores across each cluster, thereby facilitating benchmarking and longitudinal tracking.

### 2.4. Statistical Analysis

Analyses were performed using Stata 15.1 (Stata Corp., College Station, TX, USA).

### 2.5. Ethical Considerations

This study was approved by the Institutional Review Board of Taipei Veterans General Hospital, Taiwan (2017-07-015CC#1).

## 3. Results

### 3.1. Respondent Characteristics 

As Table 1 depicts, the dataset contains 45,242 questionnaires collected from 200 Taiwan hospitals. We dropped 8079 questionnaires that were missing responses for the hospital-level variables because we had built a multi-group model by hospital level. Eventually, 37,163 questionnaire survey results were analyzed. It was unsurprising that 70.6% of the respondents were nurses because they comprised the majority of the hospital workforce.

### 3.2. PA Limitations 

Taking the TC domain as an example, PA scores of all hospitals in the dataset were calculated and presented on a typical PA plot (top-left pane, Figure 1). There were three hospitals with the same PA value of 53.9% (Hospital A, B, and C). We presented the distribution of respondent-level scores in each cluster with a KDE plot (Figure 1). Distributions of individual-level safety attitudes for these hospitals differed in central tendency, dispersion, and shape.

### 3.3. The “Safety Culture Grid” Plot

To overcome the limitations of PA, we developed a new plot layout conveying more information. This was named “Jeong & Lee plot” following developers’ names (Figure 2). Each dot denotes a cluster (hospital). The y-axis depicts the median score of a hospital, and the x-axis describes the horizontally flipped IQR of the hospital. Medians (y-axis) span from approximately 17 to 93, and IQRs (x-axis) range from 13 to 82. The value of the safety grid plot emerges in its mapping of hospital safety culture, utilizing two types of information—central tendency and spread—simultaneously. Thus, the location of each dot can be denoted as a two-dimensional coordinate, as shown in parentheses (Figure 2). 

For illustrative purposes, we chose hospital X and Y from each of the two quadrants (right upper and left upper quadrant). Hospital X shows a high median value and small IQR (19, 79), meaning that most respondents of X express a quite high TC level and thus the distribution of scores is relatively tight around the median. If we used a single value, such as PA or mean to describe a hospital [15,16,17], our verdict on the comparison between X and Y might be “similar enough”. However, with two value coordinates mapped, we could see that X and Y significantly differ in safety culture status.

## 4. Discussion

This study identified the limitations of widespread PA scoring schemes and their presentation methods. Modified analytic and presentation techniques that apply hospitals’ median and IQR scores to each SAQ-C domain can provide more information for frontline safety managers and policy makers than that currently available. 

Since 2009, hospital safety culture surveys have been administered by the JCT. Hospitals participating in the study receive feedback consisting of PA scores for each domain. JCT also provides the de-identified average PA score of all participating hospitals. As a result, each hospital can examine its performance and compare it to the mean for its peers. Survey results thus help hospitals to determine the appropriate direction of patient safety programs. As hospitals regularly participate in the survey, they can also examine the effect of safety initiatives by noting changes in PA scores across years. Therefore, precision of measurement results and other information emerging from the SAQ-C survey is essential for hospitals to implement safety culture improvement tailored to their needs. Given that the PA scoring scheme has been used for ten years in Taiwan, our research suggests a need for modification not only of the SAQ-C instrument but its related analytic and explanatory methodology [18,19,20,21,22].

Since its development, PA has served as the primary metric for SAQ [9]. However, for survey respondents working at the same unit or hospital, it is a single number with limited information about safety culture topography. Our research suggests that hospital safety managers will benefit from additional survey feedback that is particularly valuable when they initiate projects tailored to their hospital’s unique situation. Taking three hospitals with the same PA score regarding TC as an example (Figure 1), Hospital A, had a higher proportion of respondents who “agree slightly” and “agree strongly”. Thus, safety programs need to focus more on employees who answered “neutral” and “disagree slightly” concerning teamwork. For Hospital B, respondents with “neutral” and “agree slightly” regarding TC dominated, with relatively few answering “disagree slightly” and “agree strongly”. Safety programs may seek to focus on these employees. 

In Hospital C, more respondents answered “disagree strongly” and “disagree slightly” than the other two hospitals. Managers may wish to focus safety improvement projects on these employees, rather than those who responded “agree strongly” and “agree slightly”. In all cases, current methodology of assigning a simple mean score or PA score does not provide such comprehensive and valuable information to participants.

Hospitals that continuously participate in the safety culture survey may realize more benefits from the utilization of median and IQR scores than that from the PA score alone. Over the past decade, many Taiwan hospitals have participated in the TPSC survey. They can compare changes in mean PA scores of each SAQ-C domain across years. If a longitudinal comparison indicates that mean PA scores do not significantly change, managers and frontline employees may question the performance of the safety improvement programs. However, after introduction of the methodology proposed here, both central tendency and spread can be tracked and examined across years. Using this information, hospitals can design teamwork improvement measures early in safety culture program development. Such measures will facilitate enhancement of the group mean concerning safety attitudes. Programs focusing on specific groups of employees can also be initiated to address the aspects of safety culture that need improvement. As these are implemented, the IQR range will narrow though the median score may appear similar across years. The performance of safety culture improvement programs can thus be measured and examined using two parameters.

## 5. Conclusions

More than ten years have passed since Taiwan launched the national comprehensive patient safety culture survey. This study demonstrates that a new scoring methodology can provide safety managers with more valuable information than the current approach, which will help them better understand both the hospital’s safety culture and the performance of safety improvement programs. The strength of the Jeong & Lee plot is that it shows both the safety-related goals of a hospital and the path to them through dissemination of practices that improve patient safety. This concept empowers hospitals and managers by providing both a cross-sectional benchmark and longitudinal tracking of cultural topography as indicated through cluster change. 

## Figures and Tables

**Figure 1 ijerph-16-03624-f001:**
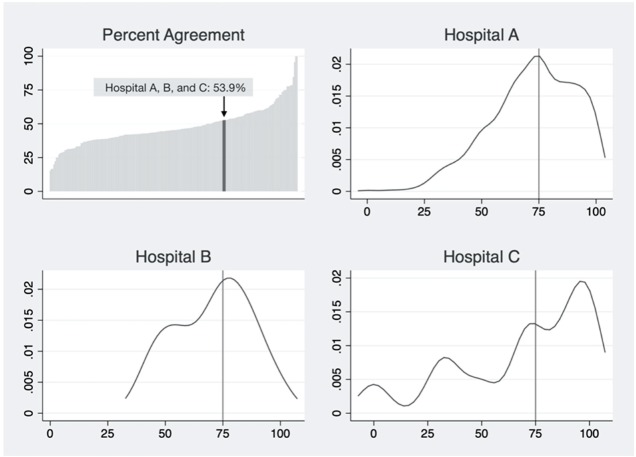
Percentage agreement graph and KDE plots of three individual hospitals where their teamwork climate PA are the same. Note: Each of the panels, Hospitals A, B, C, reflects a kernel density estimate (KDE) plot where the x-axis denotes 0–100 scores from a typical method of individual score calculation, and the y-axis is density. Lines going beyond 100 or below 0 are the result of bandwidth for smoothing.

**Figure 2 ijerph-16-03624-f002:**
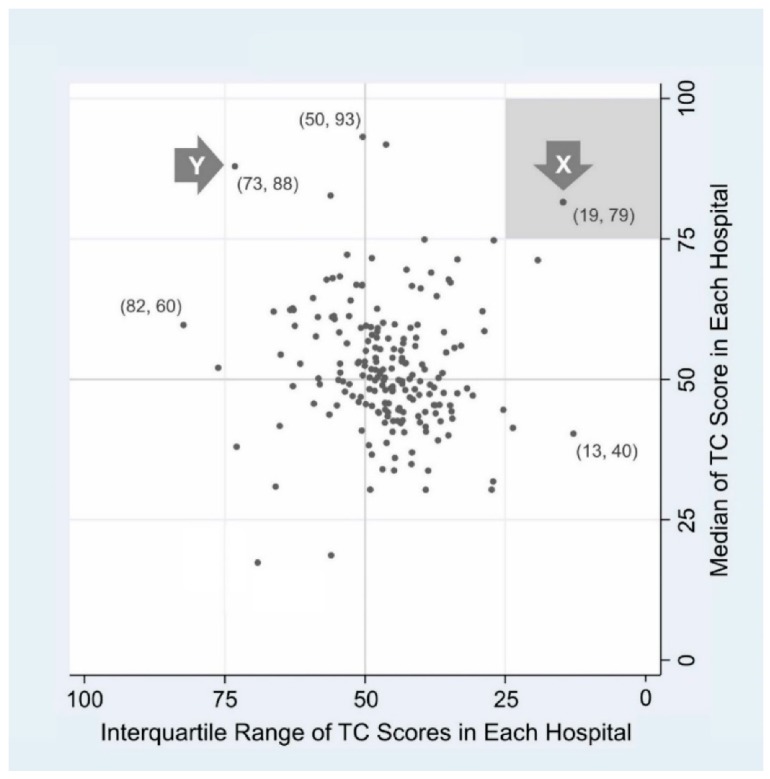
The Jeong & Lee plot of 200 hospitals participating in the Taiwan patient safety culture survey. Note that the method we proposed can be applied to any safety culture survey instrument. Because of space limits, we rounded off to the nearest whole number. Usually work area is used as an analytic cluster, but in this national sample, we used hospital as the analytic cluster. Users can always choose and switch group levels as unit of analysis.

**Table 1 ijerph-16-03624-t001:** Characteristics of respondents for a nationwide hospital patient safety culture survey in Taiwan.

Characteristics	No. of Respondents	%
Gender		
Male	4375	11.8
Female	32,788	88.2
Age Group (years old)		
≦20	110	0.3
21–30	19,668	55.5
31–40	11,656	31.7
41–50	4422	12.0
51–60	829	2.3
>60	58	0.2
Job type		
Doctors	2369	6.4
Nurses	26,229	70.6
Technicians	3054	8.2
Pharmacists	1835	4.9
Administrative Staff	792	2.1
Others	806	2.2
Missing	2078	5.6
Hospital Level (No. of Hospital)		
Medical Centers (20)	16,613	44.7
Regional Hospitals (57)	13,510	36.4
District Hospitals (104)	5698	15.3
Psychiatric Hospital (19)	1342	3.6
Total	37,163	100.0

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
