# Peer review of "The Hospital Patient Safety Culture Survey: Reform of Analysis and Visualization Methods"

_ijerph, 2019, doi:10.3390/ijerph16193624_

Round 1
Reviewer 1 Report
Please see attached comments.

Author Response
Dear Reviewer,
Appreciate for your great comments and please see the attached file.
Best regards,
Dr. Lee

Reviewer 2 Report
As mentioned also in the Abstract, the results would suggest “how to reform the SAQ analysis method” – it’s unclear whether this regards the verification of the psychometric adequacy of the instrument or its scoring in everyday practice. This is also relevant for another point: there is no discussion of the results concerning the practical use of the results, apart from the psychometric check (through the MIRT approach) at the overall level of a country. What are the users of the SAQ advised to change in their scoring of the instrument, as implied by the first section of the results (using the MIRT approach)? The same statement – “This study showed how an IRT-based approach could serve as a foundation of SAQ analysis reform” appears in Conclusions (line 245), but the nature of this reform – apart from using the median and the interquartile range instead of the PA – is lacking.
Moreover, if the future SAQ users are advised to use MIRT in their analysis, the benefits of this switch in terms of the objective of the SAQ administration should be better explained. The SAQ is a measure of patient safety culture, thus aiming to offer an overall index of employee perceptions on its dimensions, and not an exact measurement of each individual. But the main benefits of the MIRT approach, at least as described now (lines 222-234), concern precision in the sense of diagnosing the exact level of the “trait” possessed by an individual, in the classical psychodiagnosis sense. For instance, the paper states that “if a high TC level is not measured precisely, we can add an item focusing on the high trait level to equalize precision throughout the trait level” (line 228); yet, I doubt that managers / authorities would be so interested in measuring the exact safety culture perceptions of the health system employees so that to invest such effort in the individual tailored administration of the questionnaire. This again highlights the need to better explained the benefits of the approach used and of the results in the particular context of the SAQ regular use.
Generally, the logic of the statistical procedures needs to be better explained in order for the readers of the IJERPH who are less specialized in data analysis to fully understand the work done and to get a grasp of the potential application of this study.
In particular, more explanation is needed concerning c1-c4: (line 111) “Four intercepts indirectly indicate the locations of the four switching points” – since c1-c4 are among the main item parameters estimated, and since they indicate (line 161) “that a Likert scale cannot be treated as a linear interval scale with SAQ-C” more details are needed concerning the significance of these intercepts and the actual values in Table 2 that support this assertion.
Other specific comments:
- line 60 “Although empirical Bayes or high-order models have been applied to SAQ data to reduce noise in data and improve for better precision [21, 22], more comprehensive researches needed” – this should better argued, by describing the flaws of the previous attempts.
- line 217 “Both approaches passed the GOF tests, but MIRT has an approximately 0.1 higher factor loading for each item” – it’s unclear whether this refers to the results of another study
- line 166 “Finally, all the correlation coefficients between domains were high” – these correlations should also be reported in a table.
There are some wording errors that need fixing:
- line 58 “To truly utilizing…”
- line 62 “more comprehensive researches needed”
- Table 2 SC “Perceiption”
- line 204 “In this manner, Y is likely to show a left-skewed distribution” – I find it unclear the use of “in this manner” here
Author Response
Dear Reviewer 2,
Please see the attached file.
Dr. Wui-Chiang Lee

Reviewer 3 Report
I would recommend authors the following:
1) Please consider editing for grammar and word usages. There are paragraphs that appear disjointed or repetitive.
2)While it would have been more interesting to group results by providers, this research by Jeong et al. is focused on developing new methodolgy to interpret response from Likert scale survey (Taiwan's Safety Attitude Questionnaire).
Author Response
Dear Reviewer,
Thanks for your comments and please see the attached file.
Best Regards,
Dr. Lee

Round 2
Reviewer 2 Report
Thank you for addressing my previous comments.